# Socioeconomic Costs of Dementia Based on Utilization of Health Care and Long-Term-Care Services: A Retrospective Cohort Study

**DOI:** 10.3390/ijerph18020376

**Published:** 2021-01-06

**Authors:** Eun-Jeong Han, JungSuk Lee, Eunhee Cho, Hyejin Kim

**Affiliations:** 1Health Insurance Research Institute, National Health Insurance Service, Wonju 26464, Korea; 070007@nhis.or.kr (E.-J.H.); fjslee@nhis.or.kr (J.L.); 2Mo-Im Kim Nursing Research Institute, Yonsei University College of Nursing, Seoul 03722, Korea; ehcho@yuhs.ac; 3Red Cross College of Nursing, Chung-Ang University, Seoul 06974, Korea

**Keywords:** cohort studies, cost of illness, dementia, long-term care, long-term care insurance

## Abstract

This study examined the socioeconomic costs of dementia based on the utilization of healthcare and long-term care services in South Korea. Using 2016 data from two national insurance databases and a survey study, persons with dementia were categorized into six groups based on healthcare and long-term care services used: long-term care insurance users with home- and community-based services (n = 93,346), nursing home services (n = 69,895), and combined services (n = 16,068); and long-term care insurance non-users cared for by family at home (n = 192,713), living alone (n = 19,526), and admitted to long-term-care hospitals (n = 65,976). Their direct and indirect costs were estimated. The total socioeconomic cost of dementia was an estimated US$10.9 billion for 457,524 participants in 2016 (US$23,877 per person). Among the six groups, the annual per-person socioeconomic cost of dementia was lowest for long-term care insurance users who received home- and community-based services (US$21,391). It was highest for long-term care insurance non-users admitted to long-term care hospitals (US$26,978). Effective strategies are necessary to promote long-term care insurance with home- and community-based services to enable persons with dementia to remain in their communities as long as possible while receiving cost-efficient, quality care.

## 1. Introduction

Older adults living with dementia accounted for 10% (705,473) of the total older adult population (65 years or older) in South Korea in 2018, and this proportion is anticipated to rise to 16% (3,030,000) in 2050 [1]. The continuing growth of the population with dementia is a worldwide phenomenon [2]. With disease progression, persons with dementia become more dependent on others for activities of daily living (ADL) and experience diverse health problems that require prolonged medical attention. Caregivers of these individuals (mostly family) also experience tremendous difficulty and feel burdened emotionally and financially [3]. In particular, the cost of care is one of the critical concerns for persons with dementia, their families, and the government [1,3].

The costs of dementia—estimates of its socioeconomic impact—refer to all costs of dementia care incurred by patients, their families and society. The costs include direct medical, direct social (paid and professional home care and residential/nursing home care), and informal (unpaid) care costs [2,4]. Costs vary worldwide, depending on types of national health and social care systems, availability of resources, national economic status, wage levels, and proportion of unpaid, often un-recognized, informal caregiving [4]. Estimating these costs helps elucidate the impact of national health and social care systems and services on the economic burden to the society and facilitate discussions regarding cost-efficient policies and services and proper allocation of healthcare and social resources.

The global socioeconomic costs of dementia are rapidly increasing—there was a 35.4% increase between 2010 and 2015, and the figure will reach approximately US$2 trillion by 2030 [2,4]. For example, North America experienced a 26.3% increase from US$213 billion in 2010 to US$269 billion in 2015; Western European countries noted similar costs and trends. East Asian and Pacific countries experienced a 46.2% increase from US$105 billion in 2010 to US$153 billion in 2015 [4]. In South Korea, one of the East Asian and Pacific countries, the total annual socioeconomic cost of dementia was anticipated to increase from US$11 billion in 2015 to US$28.6 billion in 2030 [5]. These increasing costs indicate an increasing socioeconomic burden on persons with dementia, their families, and the government.

Recently, the South Korean government has highlighted a national responsibility for dementia care [1]. South Korea has two main national insurance programs that can provide financial support for dementia care. One is the National Health Insurance (NHI), which covers costs of medical services (e.g., hospitalization, including long-term care [LTC] hospitals, outpatient visits, medications) and has 97.2% of the population as beneficiaries [6]. When persons with dementia receive medical care, the NHI bears part of the expenses. The other program is the LTC Insurance (LTCI) for older adults, a social insurance system that commenced in 2008 to address the increasing LTC needs of older adults and their families [7]. This LTCI system is comparable to those of Germany and Japan [8]. Applicants for this insurance must be adults aged 65 and older; individuals aged under 65 who wish to apply must have geriatric diseases, such as dementia, stroke, and Parkinson’s disease. Once an application is submitted by patients, families, or others, NHI service employees conduct an assessment to determine if the applicant is qualified for LTCI. In 2018, 8.4% of older adults were receiving LTCI, and the proportion has been increasing gradually [7]. LTCI provides beneficiaries and their families with three types of insurance benefits—home- and community-based care (visiting care, bathing, nursing, day-and-night care center, short-term respite care, and equipment service), institutional care (nursing home care), and special cash—based on beneficiaries’ or families’ preferences [7]. Different from NHI, the LTCI focuses on services supporting beneficiaries’ physical or housework activities, rather than on medical services (e.g., hospitalization, outpatient visits). Additionally, LTCI prioritizes home- and community-based care for older adults with difficulties in ADL and their families [9]. Furthermore, the government modified the existing LTCI levels (level 1 [severely dependent] to 3 [mildly dependent]) from three to five and initiated a dementia-specific level in 2014 to ensure that more adults with dementia and their families benefit from LTCI [10].

The socioeconomic costs of dementia may differ depending on the type of healthcare and LTC services used by these individuals and their families. Accordingly, this study examined the socioeconomic costs of dementia based on the utilization of healthcare and LTC services in South Korea. The findings provide information regarding the most cost-efficient service type for dementia care and contribute to developing practical strategies to reduce financial burden while providing quality care.

## 2. Materials and Methods

### 2.1. Data Sources

We used 2016 data retrieved from three sources in South Korea: (a) NHI database, (b) LTCI database, and (c) a study entitled “Survey of Family Caregiving for LTCI Beneficiaries” (hereafter “Survey of Family Caregiving”) [11]. The national-level NHI database includes data on beneficiaries’ demographics, health status, and healthcare utilization [12]. The LTCI database, another national-level data source, includes applicants’ assessment data (e.g., demographic information, health and functional status, final LTC benefit level) and beneficiaries’ LTC service utilization [13]. The Survey of Family Caregiving study was conducted by Han and colleagues between 10 June and 20 July 2016, to examine the caregiving status of applicants who were approved for LTCI benefits [11]. A total of 2625 family caregivers of those applicants completed a nine-domain survey: sociodemographic information, caregiving status, caregiving expenses, stress management, health status and risk behaviors, caregiving burden, depressive symptoms, self-efficacy, and quality of life. The “caregiving expenses” domain asked about expenses associated with medical services, LTC services, transportation, foods, diapers, and paid care workers during the previous month [11]. Of the completed surveys, we used data from 891 family caregivers of persons with dementia (523 used home- and community-based services, 143 used nursing home services, and 225 did not use LTCI benefits). Individuals who did not use the LTCI benefits (n = 225) were cared for by family at home (n = 109, 48.4%), admitted to LTC hospitals (n = 105, 46.7%), or lived alone (n = 11, 4.9%). Our secondary analyses of existing data were approved by the institutional review board of the first author’s institution and qualified for exemption. 

### 2.2. Subjects

From the NHI database, we identified adults who had a record of South Korean standard disease classification codes of dementia (i.e., F00, F01, F03, F10.7, G30) and prescriptions of at least one dementia-related medication from 1 January to 31 December 2016 (N = 457,524). In South Korea, dementia is diagnosed through three procedural steps: a screening test for cognitive impairment, diagnostic tests to confirm dementia by psychiatrists and neurologists, and differential tests to identify causes of dementia [14]. Of this total, 95.1% were aged 65 years and older, while 4.9% were under 65 years. Further, 71.2% were women and 28.8% were men; the proportion of women was much higher than that in the population aged 65 years and older (58% vs. 42%) and slightly lower than that in the population aged 85 years and older (75% vs. 25%) [15]. These persons with dementia were categorized into LTCI users (n = 179,309, 39.2%) and non-users (n = 278,215, 60.8%). LTCI users were further divided into three groups based on their LTC service plans: (a) home- and community-based services (n = 93,346, 20.4%), (b) nursing home services (n = 69,895, 15.3%), and (c) combined services (n = 16,068, 3.5%). LTCI non-users were also sub-categorized into three groups based on the healthcare services they received: (d) those cared for by family at home (n = 192,713, 42.1%), (e) those living alone (n = 19,526, 4.3%), and (f) those admitted to LTC hospitals (n = 65,976, 14.4%). LTCI non-users admitted to LTC hospitals included only those with a long-term stay. We defined long-term hospitalization as receiving care in LTC hospitals for at least 222 days, which was the mean length of stay of all patients with admissions to LTC hospitals in 2016. 

### 2.3. Measures: Items of Socioeconomic Costs

We classified the socioeconomic costs of dementia using Kang et al.’s [16] comprehensive approach, which accounts for family caregivers’ productivity loss in these estimations. We also used both top-down (calculating costs of care attributable to dementia care using national-level databases) and bottom-up (extrapolating costs of care for a sample of persons with dementia to all persons with dementia) approaches to estimate the socioeconomic costs of dementia [17]. Table 1 presents key items, definitions, and data sources used to estimate the cost of each item.

These costs included both direct and indirect costs. Items constituting direct costs were direct medical and non-medical costs. Direct medical costs refer to expenditures on medical services, such as hospitalization, outpatient visits, and medications, paid by NHI and the patient. Data pertaining to direct medical costs were obtained from the NHI database (for NHI payment) and the Survey of Family Caregiving study (for patient payment). Direct non-medical costs were expenditures on LTC services, such as home- and community-based care and nursing home services, which were borne by LTCI and patients. These data were obtained from the LTCI database (for LTCI payment) and the Survey of Family Caregiving study (for patient payment).

Indirect costs comprised two items: general indirect costs and family caregivers’ productivity loss. General indirect costs were expenditures on healthcare-related transportation, foods, diapers, and paid care workers. We used data from the Survey of Family Caregiving study to estimate general indirect costs. Caregiver productivity loss was defined as income loss because of absence from work as a result of taking care of the family member with dementia, and calculated using data from the NHI database and the Survey of Family Caregiving study.

### 2.4. Estimation of Socioeconomic Costs

Annual socioeconomic costs were estimated at the person and group levels. We calculated the average annual direct costs paid by NHI and LTCI in each patient group, using the NHI and LTCI data. To estimate per-person annual direct costs paid by patient and general indirect costs, we used data from the Survey of Family Caregiving study: we multiplied the average monthly out-of-pocket expenses on medical and LTC services, transportation, foods, diapers, and paid care workers in each group by 12. Given that the Survey of Family Caregiving study data has no “combined service” group of LTCI users, we applied the average expenses on these items in the other two groups of LTCI users to the group of LTCI users with combined services. The caregiver productivity loss per person was calculated [18] by adding hospitalization days and a third of the outpatient visit days and then multiplying the sum by the employment rate by gender and age (obtained from the economically active population survey of Statistics Korea [19]) and daily average wage by gender and age (obtained from the national survey report on labor conditions by employment type [20]). The sum of per-person annual direct and indirect costs in each patient group was multiplied by the size of the patient group to estimate per-group socioeconomic costs.

## 3. Results

### 3.1. Total Socioeconomic Costs

The 2016 total socioeconomic cost of dementia for 457,524 persons with dementia was estimated to be US$10.9 billion (the average exchange rate in 2016—1200 Korean won/US$—is applied throughout) with US$6.2 billion (56.7%) for direct costs and US$4.7 (43.3%) for indirect costs, yielding an average of US$23,877 per person with dementia (see Table 2).

Government coverage (NHI and LTCI) accounted for 37.8% of the total annual socioeconomic cost. Overall, LTCI users had lower annual socioeconomic costs per person than did LTCI non-users (US$22,031 vs. US$25,068). Specifically, an LTCI non-user admitted to an LTC hospital had the highest socioeconomic cost (US$26,978), followed by an LTCI non-user cared for by family (US$24,765). Conversely, the group of LTCI users receiving home- and community-based services marked the lowest socioeconomic cost per person (US$21,391).

### 3.2. Direct Costs

The total annual direct cost per person was estimated to be US$13,537, with 66.6% paid by the government and 33.4% paid by persons with dementia (see Table 3). Per-person direct cost was highest in LTCI non-users admitted to LTC hospitals (US$26,978), followed by LTCI users receiving nursing home services (US$19,822). Conversely, LTCI non-users living alone had the lowest per-person direct cost (US$5825).

Per-person direct medical cost was highest in LTCI non-users admitted to LTC hospitals (US$26,978) among the six groups of persons with dementia. In this group, the direct medical cost paid by the NHI was US$17,346 (64.3%) and that paid by a patient (out-of-pocket spending) was US$9632 (35.7%). Conversely, at US$4976, the lowest per-person direct medical cost was associated with LTCI users receiving nursing home services (US$3626 [72.9%] paid by NHI and US$1350 [27.1%] paid by a patient).

Direct non-medical cost per person was highest in LTCI users receiving nursing home services (US$14,846) and lowest in those receiving home- and community-based services (US$6915) among the three LTCI user groups. In LTCI users receiving nursing home services, non-medical costs paid by LTCI amounted to US$9456 (63.7%), while the costs paid by a patient amounted to US$5390 (36.3%).

### 3.3. Indirect Costs

The 2016 total indirect cost per person was estimated to be US$10,340, with 75.7% for family caregiver productivity loss and 24.3% for general indirect cost (see Table 4). LTCI non-users cared for by family at home had the highest per-person indirect cost (US$18,143), whereas LTCI users receiving nursing home services had the lowest cost (US$2918).

General indirect cost (expenditures on transportation, foods, diapers, and paid care workers) per person was similar between LTCI users receiving home- and community-based services (US$3715) and LTCI non-users cared for by family (US$3706). For LTCI users receiving nursing home services and LTCI non-users admitted to LTC hospitals, general indirect cost was estimated to be zero because that cost was included in their hospitalization or nursing home fees.

Caregiver productivity loss per person with dementia was more than four times higher in LTCI non-users cared for by family (US$14,437) and those living alone (US$12,875) than in LTCI users receiving home- and community-based services (US$3175). LTCI users receiving nursing home services also exhibited caregiver productivity loss of US$2918 owing to their need to be accompanied to hospitals and outpatient clinics for medical services. Conversely, caregiver productivity loss was assumed to be zero for LTCI non-users admitted to LTC hospitals. Since persons with dementia in LTC hospitals stayed there for a longer period and could receive medical services on site, we assumed that their family members did not need to take leave of absence from work.

## 4. Discussion

This study examined the socioeconomic costs of dementia in South Korea by comparing six groups of persons with dementia based on the utilization of healthcare and LTC services. In 2016, the total socioeconomic cost of dementia was estimated to be US$10.9 billion for 457,524 persons with dementia, which represents US$23,877 per person. Among six groups of persons with dementia, the annual socioeconomic cost of dementia per person was lowest in LTCI users who received home- and community-based services (US$21,390.58) and highest in LTCI non-users admitted to LTC hospitals (US$26,977.89). These findings support the South Korean government’s promotion of LTCI with home- and community-based services [9] and the international trend toward home- and community-based dementia care [21].

The annual socioeconomic cost of dementia per person in our study was slightly higher than that estimated for G20 countries in 2015 (US$20,187) but much lower than that estimated for G7 countries (US$ 43,680) [4]. Moreover, compared with Kang et al.’s study [16], the number of NHI beneficiaries with dementia was 5.4 times higher (457,524 vs. 85,281), and the annual per-person socioeconomic cost of dementia was 3.6 times higher (US$23,877 vs. US$6650). Possible reasons for the growth in the number of NHI beneficiaries with dementia include the growing geriatric population, increased early dementia screening, and people’s increased awareness and knowledge of dementia [1,22]. The increased per-person socioeconomic costs of dementia may also be attributed to increased healthcare utilization, supply prices, and care workers’ fees [23,24].

Frequent or long-term hospitalization places a major financial burden on the government, persons with dementia, and their families [25,26]. In our study, LTCI non-users who were admitted to LTC hospitals throughout 2016 presented the highest socioeconomic cost of dementia per person (US$26,977.89). As the indirect cost was estimated to be zero in this group, the hidden cost might impose an additional burden on persons with dementia and their families. The LTC hospital is a venue in which debilitated persons receive medical treatments in addition to LTC services for the purpose of cure and recuperation [27]. This is different from nursing homes, where the main focus of care is to provide support in ADL. Despite the high costs of LTC hospitalization, many persons with illnesses and their families (including those approved for LTCI benefits) preferred LTC hospitals to LTCI’s nursing home services. In a recent study, the factors associated with family caregivers being more likely to select LTC hospital services than LTCI with nursing home or home- and community-based services included the following: lower patient age, higher family income, a greater number of comorbidities, and poorer psychiatric health of caregivers [28]. Considering the high costs of LTC hospitalization, strategies should be developed to have persons with dementia remain in the community as long as possible while receiving quality care that meets both patients’ and families’ needs. If this is not possible, there should be a focus on reducing the length of stay in LTC hospitals.

Our findings support that an LTC-related social security system, like LTCI, helps reduce the financial burden of caring for these individuals. In particular, LTCI’s home- and community-based services were the least costly care modality among the six groups of persons with dementia. This finding is consistent with existing evidence [29]. Furthermore, home- and community-based care is associated with positive patient outcomes. In South Korea, Lee et al. [30] found that LTCI beneficiaries using home- and community-based services had better cognitive and physical functions than did those utilizing nursing home services after two years, adjusting for covariates (e.g., demographics, comorbidities). Similarly, in Japan, LTCI beneficiaries who used home- and community-based services, such as respite care and rental services for assistive devices, experienced less institutionalization and hospitalization than those who did not [31]. This care modality is a potentially cost-efficient approach in dementia care, as well-coordinated home- and community-based services may delay or reduce institutionalization and hospitalization [32].

However, home- and community-based services for persons with dementia require a considerable amount of involvement and responsibility from family caregivers. In a recent study, informal care by family caregivers was defined as “*hours of informal support provided to the person with dementia in an average day in respect of each type of support: ADL, instrumental ADL, and supervision*” ([29], p. 1178). In Jutkowitz et al. [33], the caregiving duration was 151 h a month at the diagnosis of dementia, which increased to 283 h a month in eight years. The cost of informal care is calculated using various methods; in one study, estimations included opportunity cost using the “*average hourly wage for all industrial sectors in Ireland*” and replacement cost using the “*market wage for a healthcare assistant*” ([29], p. 1179). Among community-dwelling persons with dementia, this informal care is a main driver of socioeconomic costs, accounting for 60–84% of the total costs of dementia [34].

In our study, we estimated the costs of family caregivers’ productivity loss as part of informal care. Among community-dwelling persons with dementia, the cost per person was highest in LTCI non-users cared for by family at home (US$14,437, 58.3% of the total cost of dementia) and lowest in LTCI users who received home- and community-based services (US$3175, 14.8% of the total cost of dementia). However, these costs were much higher than those for LTCI users receiving nursing home services and LTCI non-users admitted to LTC hospitals. These findings suggest that LTCI with home- and community-based services partly relieves family caregivers’ burden, but this care modality still requires systematic support for informal caregiving.

To improve LTCI with home- and community-based services, the South Korean government launched an integrated home- and community-based service, which employs a case management approach, in August 2019 [35]. Before the implementation of the integrated service, beneficiaries and their families chose from various home- and community-based services and independently contacted institutions that offered the desired services [35]. Consequently, approximately 75% of beneficiaries using LTCI’s home- and community-based services received only one service, and those who used visiting nursing services accounted for only 4% [36]. Conversely, the integrated home- and community-based service offers an individualized bundle of services with enhanced nursing services. Any bundle should include at least weekly nursing services at home or daycare centers; these services are provided by interdisciplinary teams. Such a case management approach tends to reduce the institutionalization of persons with dementia, depression in caregivers, and total costs of services [37]. Further research is needed to examine the cost-effectiveness of the integrated home- and community-based service for dementia care.

This study has some limitations. First, although the severity of dementia and number of comorbidities are associated with an increase in socioeconomic costs of dementia [38,39], we did not consider these factors. Second, there was a possibility that the cost of family caregivers’ productivity loss for LTCI non-users admitted to LTC hospitals was higher than zero. Finally, while including family caregivers’ productivity loss in estimating the socioeconomic cost of dementia, we did not quantify their unpaid caregiving in daily care. Despite these limitations, this study presents valuable information regarding the socioeconomic costs of dementia based on the utilization of healthcare and LTC services in South Korea. Additionally, it provides evidence to support a social insurance system for LTC and its home- and community-based services as a potentially cost-efficient approach for dementia care.

## 5. Conclusions

Based on our findings, LTCI with home- and community-based services was the least costly dementia care approach among the six healthcare and LTC service types. Future research needs to consider dementia stages, comorbidities, family members’ health, and unpaid caregiving to estimate more comprehensively the socioeconomic costs of dementia. Moreover, studies that develop evidence-based, innovative strategies (e.g., policy, educational programs, information technologies) and examine their cost-effectiveness are necessary to promote LTCI with home- and community-based services and provide practical and systematic support to family caregivers. These efforts may eventually enable persons with dementia to remain in their communities for as long as possible while receiving quality care.

## Figures and Tables

**Table 1 ijerph-18-00376-t001:** Measure: Key items, Definitions, and Data Sources.

Domain		Items		Definition	Data Sources
Direct Costs	Medical costs	NHI payment	Insurer payment	Annual medical costs paid by NHI (hospitalization, outpatient visit, and medication)	NHI database
			Patient payment	Annual medical costs paid by a patient (hospitalization, outpatient visit, and medication)	Survey of Family Caregiving
	Non-medical costs	LTC payment	Insurer payment	Annual LTC costs paid by LTCI (nursing home service, home care service, or both services)	LTCI database
			Patient payment	Annual LTC costs paid by a patient (nursing home service, home care service, or both services)	Survey of Family Caregiving
Indirect Costs	General indirect costs	Transportation costs		Annual expenditures on transportation for hospital or outpatient visits and LTC service use of a patient with dementia	Survey of Family Caregiving
		Food costs		Annual expenditures on food consumed by a patient with dementia	
		Diaper costs		Annual expenditures on diapers used by a patient with dementia	
		Care-worker costs		Annual expenditures on paid care workers to take care of a patient with dementia	
	Productive loss (or income loss) ^1^	Caregiver income loss		Caregiver’s income loss related to a leave of absence from work to take a patient to hospitals or other medical centers	NHI database Survey of Family Caregiving

^1^*Formula*: (admission days + outpatient visit days/3) × (employment rate by sex & age) × (daily average wage by sex & gender). LTC = long-term-care; LTCI: long-term-care insurance; NHI: national health insurance.

**Table 2 ijerph-18-00376-t002:** Total Socioeconomic Costs of Dementia.

Patient Group by Service Type		N	Direct Costs US$ (%)		Indirect Costs US$ (%)		Total Costs US$ (%)	
			Per Person	Per Group	Per Person	Per Group	Per Person	Per Group
LTCI Users	HC service	93,346	14,501	1,353,540,803	6890	643,184,433	21,391	1,996,725,236
			(67.8)	(67.8)	(32.2)	(32.2)	(100.0)	(100.0)
	NH service	69,895	19,822	1,385,500,802	2918	203,961,065	22,741	1,589,461,867
			(87.2)	(87.2)	(12.8)	(12.8)	(100.0)	(100.0)
	Combined (HC+NH)	16,068	17,741	285,053,216	4904	78,800,953	22,645	363,854,169
			(78.3)	(78.3)	(21.7)	(21.7)	(100.0)	(100.0)
	Subtotal	179,309	16,866	3,024,094,821	5165	925,946,452	22,031	3,950,041,273
			(76.6)	(76.6)	(23.4)	(23.4)	(100.0)	(100.0)
LTCI Non-users	Cared for by family at home	192,713	6622	1,276,173,429	18,143	3,496,326,115	24,765	4,772,499,545
			(26.7)	(26.7)	(73.3)	(73.3)	(100.0)	(100.0)
	Living alone	19,526	5825	113,747,558	15,785	308,215,274	21,610	421,962,832
			(27.0)	(27.0)	(73.0)	(73.0)	(100.0)	(100.0)
	Admitted to LTC hospitals	65,976	26,978	1,779,892,941	0	0	26,978	1,779,892,941
			(100.0)	(100.0)			(100.0)	(100.0)
	Subtotal	278,215	11,393	3,169,813,928	13,675	3,804,541,389	25,068	6,974,355,317
			(45.4)	(45.4)	(54.6)	(54.6)	(100.0)	(100.0)
Total		457,524	13,537	6,193,908,749	10,340	4,730,487,841	23,877	10,924,396,590
			(56.7)	(56.7)	(43.3)	(43.3)	(100.0)	(100.0)
Government’s payment (%)				66.6 ^1^		0.0		37.8 ^2^

^1^ Proportion of the government (NHI and LTCI)’s payment to direct cost. ^2^ Proportion of the government’s payment to total socioeconomic cost. HC = home- and community-based; LTC = long-term-care; LTCI = long-term-care insurance; NH = nursing home; NHI = national health insurance.

**Table 3 ijerph-18-00376-t003:** Per-Person Direct Costs.

Patient Group by Service Type		N	Direct Costs (Per Person) US$ (%)				
			Subtotal	Medical Costs		Non-Medical Costs	
				NHI	Patient ^1^	LTCI	Patient ^1^
LTCI Users	HC service	93,346	14,501	5679	1907	4492	2423
			(100.0)	(39.2)	(13.1)	(31.0)	(16.7)
	NH service	69,895	19,822	3626	1350	9456	5390
			(100.0)	(18.3)	(6.8)	(47.7)	(27.2)
	Combined (HC+NH)	16,068	17,741	4943	1628	7263	3907
			(100.0)	(27.9)	(9.2)	(40.9)	(22.0)
	Subtotal	179,309	16,866	4813	1665	6675	3713
			(100)	(28.5)	(9.9)	(39.6)	(22.0)
LTCI Non-users	Cared for by family at home	192,713	6622	4335	2287	0	0
			(100.0)	(65.5)	(34.5)		
	Living alone	19,526	5825	4335	1490	0	0
			(100.0)	(74.4)	(25.6)		
	Admitted to LTC hospitals	65,976	26,978	17,346	9632	0	0
			(100.0)	(64.3)	(35.7)		
	Subtotal	278,215	11,393	7420	3973	0	0
			(100)	(65.1)	(34.9)		
Total		457,524	13,537	6398	3068	2616	1455
			(100.0)	(47.3)	(22.6)	(19.3)	(10.8)

^1^ Multiplying patient’s monthly payment by 12 months. HC = home- and community-based; LTC = long-term-care; LTCI = long-term-care insurance; NH = nursing home; NHI = national health insurance.

**Table 4 ijerph-18-00376-t004:** Per-Person Indirect Costs.

Patient Group by Service Type		N	Indirect Costs (Per Person) ^1^ US$ (%)					
			Subtotal	General Indirect Costs				Family Caregiver Productivity Loss
				Transpor-tation	Foods	Diapers	Paid Care Worker	
LTCI Users	HC service	93,346	6890	556	2260	564	335	3175
			(100.0)	(8.1)	(32.8)	(8.2)	(4.8)	(46.1)
	NH service	69,895	2918	0	0	0	0	2918
			(100.0)					(100.0)
	Combined (HC+NH) ^2^	16,068	4904	278	1130	282	167	3047
			(100.0)	(5.7)	(23.0)	(5.8)	(3.4)	(62.1)
	Subtotal	179,309	5165	315	1279	319	189	3063
			(100.0)	(6.0)	(24.8)	(6.2)	(3.7)	(59.3)
LTCI Non-users	Cared for by family at home	192,713	18,143	596	2278	548	284	14,437
			(100.0)	(3.3)	(12.5)	(3.0)	(1.6)	(79.6)
	Living alone	19,526	15,785	530	1800	580	0	12,875
			(100.0)	(3.3)	(11.4)	(3.7)	(0.0)	(81.6)
	Admitted to LTC hospital	65,976	0	0	0	0	0	0
	Subtotal	278,215	13,675	450	1704	420	197	10,904
			(100.0)	(3.3)	(12.5)	(3.1)	(1.4)	(79.7)
Total		457,524	10,340	397	1538	380	194	7831
			(100.0)	(3.8)	(14.9)	(3.7)	(1.9)	(75.7)

^1^ Multiplying monthly indirect costs by 12 months. ^2^ In this group, indirect costs were the average of those in HC and NH service groups. HC = home- and community-based; LTC = long-term-care; LTCI = long-term-care insurance; NH = nursing home.

## Data Availability

No new data were created or analyzed in this study. Data sharing is not applicable to this article.

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
