# Peer review of "Socioeconomic Costs of Dementia Based on Utilization of Health Care and Long-Term-Care Services: A Retrospective Cohort Study"

_ijerph, 2021, doi:10.3390/ijerph18020376_

Round 1

Reviewer 1 Report

Dear authors,

The study topic is interesting and the study is easy to read and to understand. I have somme questions and recommendations.

1) I am not sure about this sentence "The requirement of written informed consent was waived, given that we conducted a secondary analysis of existing data and did not disclose and personal identifiers." You used a survey with some family caregivers, not only a secondary analysis.

2) I would include a "Data Analysis" section in the methods to explain the performed analyses.

3) I would include the information in the discussion about future research instead of the conclusion.

Author Response

We thank the reviewers for their thoughtful comments, which enabled us to improve our manuscript (ID: ijerph-1033247). We have responded to each comment below. Changes are marked in blue in the revised manuscript.

Reviewer 1:

The study topic is interesting and the study is easy to read and to understand. I have some questions and recommendations.

1) I am not sure about this sentence "The requirement of written informed consent was waived, given that we conducted a secondary analysis of existing data and did not disclose and personal identifiers." You used a survey with some family caregivers, not only a secondary analysis.

Response: We agree that this sentence is confusing. We used only existing data. We did not conduct a survey of family caregivers for this study. Our study qualified for exemption; the IRB at the first author’s institution approved the study. Accordingly, we revised this sentence to “Our secondary analyses of existing data were approved by the institutional review board of the first author’s institution and qualified for exemption.” Please see page 3, line 105–107.

2) I would include a "Data Analysis" section in the methods to explain the performed analyses.

Response: Thank you for the suggestion. In our study, the analysis entailed primarily estimation of socioeconomic costs (total, direct, and indirect costs). Thus, we added “2.4. Estimation of Socioeconomic Costs” instead of “Data Analysis” (please see page 4). The section reads as follows:

"2.4. Estimation of Socioeconomic Costs

Annual socioeconomic costs were estimated at the person and group levels. We calculated the average annual direct costs paid by NHI and LTCI in each patient group, using the NHI and LTCI data. To estimate per-person annual direct costs paid by patient and general indirect costs, we used data from the Survey of Family Caregiving study: we multiplied the average monthly out-of-pocket expenses on medical and LTC services, transportation, foods, diapers, and paid care workers in each group by 12. Given that the Survey of Family Caregiving data has no “combined service” group of LTCI users, we applied the average expenses on these items in the other two groups of LTCI users to the group of LTCI users with combined services. The caregiver productivity loss per person was calculated [18] by adding hospitalization days and a third of the outpatient visit days and then multiplying the sum by the employment rate by gender and age (obtained from the economically active population survey of Statistics Korea [19]) and daily average wage by gender and age (obtained from the national survey report on labor conditions by employment type [20]). The sum of per-person annual direct and indirect costs in each patient group was multiplied by the size of the patient group to estimate per-group socioeconomic costs."  

3) I would include the information in the discussion about future research instead of the conclusion.

Response: Thank you for this suggestion. We, as a team, discussed the best way to include directions for future research, and decided to add such content to the Conclusion. Thus, we revised the Conclusion as follows (see page 9):

“Future research needs to consider dementia stages, comorbidities, family members’ health, and unpaid caregiving to estimate more comprehensively the socioeconomic costs of dementia. Moreover, studies that develop evidence-based, innovative strategies (e.g., policy, educational programs, information technologies) and examine their cost-effectiveness are necessary to promote LTCI with home- and community-based services and provide practical and systematic support to family caregivers. These efforts may eventually enable persons with dementia to remain in their communities for as long as possible while receiving quality care.”

Reviewer 2 Report

Thank you for sending your paper entitled “Socioeconomic Costs of Dementia Based on Utilization of Health Care and Long-Term-Care Services: A Retrospective Cohort Study” to IJERPH. After carefully review this interesting paper, the following comments are listed for your reference:

  1. Introduction (P1-2, L41-48): I suggest to include further rational for analysing the cost of dementia internationally. It would help international readers to understand the issue underneath. Are these numbers similar in the US? What about Europe? What similarities or differences do their national insurance programs have?
  2. Discussion (P7, L215-238): Please see comment above. These changes will definitely underpin the value of your findings.
  3. References (P9-10, L310-401): clustered references should not be separated (e.g. [8,9] instead of [8, 9]).
  4. English: additional proof reading would enhance it greatly. Some sentences could be restructured for clarity.

Author Response

Reviewer 2:

Thank you for sending your paper entitled “Socioeconomic Costs of Dementia Based on Utilization of Health Care and Long-Term-Care Services: A Retrospective Cohort Study” to IJERPH. After carefully review this interesting paper, the following comments are listed for your reference:

1. Introduction (P1-2, L41-48): I suggest to include further rational for analysing the cost of dementia internationally. It would help international readers to understand the issue underneath. Are these numbers similar in the US? What about Europe? What similarities or differences do their national insurance programs have?

2. Discussion (P7, L215-238): Please see comment above. These changes will definitely underpin the value of your findings.

Response to Comments 1 & 2: Thank you for the suggestions. Based on these suggestions, we have added additional rationale for international studies that estimate costs of dementia. We also added socioeconomic costs in North America, Western European, and East Asian and Pacific countries in addition to the costs in South Korea. The revised section reads as follows:

“The costs of dementia—estimates of its socioeconomic impact—refer to all costs of dementia care incurred by patients, their families, and society. The costs include direct medical, direct social (paid and professional home care and residential/nursing home care), and informal (unpaid) care costs [2,4]. Costs vary worldwide, depending on types of national health and social care systems, availability of resources, national economic status, wage levels, and proportion of unpaid, often un-recognized, informal caregiving [4]. Estimating these costs helps elucidate the impact of national health and social care systems and services on the economic burden to the society and facilitate discussions regarding cost-efficient policies and services and proper allocation of healthcare and social resources.

The global socioeconomic costs of dementia are rapidly increasing—there was a 35.4% increase between 2010 and 2015, and the figure will reach approximately US$2 trillion by 2030 [2,4]. For example, North America experienced a 26.3% increase from US$213 billion in 2010 to US$269 billion in 2015; Western European countries noted similar costs and trends. East Asian and Pacific countries experienced a 46.2% increase from US$105 billion in 2010 to US$153 billion in 2015 [4]. In South Korea, one of the East Asian and Pacific countries, the total annual socioeconomic cost of dementia was anticipated to increase from US$11 billion in 2015 to US$28.6 billion in 2030 [5]. These increasing costs indicate an increasing socioeconomic burden on persons with dementia, their families, and the government.” (see pages 1–2, lines 41-58)

The S. Korean long-term care insurance system is comparable to those in Germany and Japan. We have included a sentence in the Introduction regarding long-term care insurance. Please see page 2, line 66. 

3. References (P9-10, L310-401): clustered references should not be separated (e.g. [8,9] instead of [8, 9]).

Response: We have edited the citation style throughout the main text.

4. English: additional proof reading would enhance it greatly. Some sentences could be restructured for clarity.

Response: Thank you for the suggestion. We have reviewed the manuscript carefully and received professional editing services for the resubmission.

Reviewer 3 Report

South Korea is one of the G20 countries with high GDP and rich financial resources for looking after the elderly, including demented, citizens. Life expectancy is quite high in the country therefore the expected rise in dementia cases is higher than in most countries of the World. The national record keeping system appers very good and the Authors used these databases to conduct a retrospective study on socioeconomic costs if dementia.

The study provides insight into the care and health insurance system in South Korea and this is interesting on its own for the readership, because it appears to be a quite efficient and good system with a mix of solidarity-based sate health care and self-funding. The study concludes that people with long term care insurance and comminity based services attract the lowest socioeconomic costs and those without such insurance admitted to long-term care hospital have the highest costs. The presentation of results is correct and the discussion is balanced, including a fair account on the limitations of the study. The reference list is adequate. 

The topic is important, relevant and a report from the Korean penninsula is welcome (this paper covers only the South and the naive question arises in the Reviewer, that a comparative analysis with the North would be of interest). 

Specific questions and comments:

1) specify how was the diagnosis of dementia established? (specialist centre of what type?, GP? or other specialist?)

2) Are there 'smart home' and other information technology (IT) infrastructures (robots, video-monitoring, telemedicine, etc.) available, and how do these affect socioeconomic costs?

3) the high proportion of females to what extent reflects (or not) the proportion of women in the elderly population?

Author Response

Reviewer 3:

Specific questions and comments:

1) specify how was the diagnosis of dementia established? (specialist centre of what type?, GP? or other specialist?)

Response: We now describe how dementia was diagnosed: “In South Korea, dementia is diagnosed through three procedural steps: a screening test for cognitive impairment, diagnostic tests to confirm dementia by psychiatrists and neurologists, and differential tests to identify causes of dementia [14].” (please see page 3, lines 111–114).

2) Are there 'smart home' and other information technology (IT) infrastructures (robots, video-monitoring, telemedicine, etc.) available, and how do these affect socioeconomic costs?

Response: In South Korea, despite the recent increasing attention to smart home and information technology infrastructures, these technologies and systems are still uncommon for dementia care. Thus, smart home and information technology infrastructures rarely affected socioeconomic costs in 2016.

3) the high proportion of females to what extent reflects (or not) the proportion of women in the elderly population?

Response: “The proportion of women was much higher than that in the population aged 65 years and older (58% vs. 42%) and slightly lower than that in the population aged 85 years and older (75% vs. 25%) [15].” This content has been added to the main text (please see “2.2. Subjects” on page 3, lines 115–117).

Round 2

Reviewer 2 Report

Thank you for sending your revised paper entitled “Socioeconomic Costs of Dementia Based on Utilization of Health Care and Long-Term-Care Services: A Retrospective Cohort Study” to IJERPH and take my recommendations into account.